# Development of Activity Data for Greenhouse Gas Inventory in Settlements in South Korea

Sol-E Choi [1], Moonil Kim [2], Yowhan Son [3], Seong-Woo Jeon [3], Kyeong-Hak Lee [4], Whijin Kim [3], Sun-Jeoung Lee [1] and Woo-Kyun Lee [3,*]

1. Forest ICT Research Center, National Institute of Forest Science, 57 Hoegi-ro, Dongdaemun-gu, Seoul 02455, Republic of Korea; pine0630@gmail.com (S.-E.C.); sunjleei@korea.kr (S.-J.L.)
2. Division of ICT-Integrated Environment, Pyeongtaek University, 111 Yongyi-Dong, Pyeongtaek-si 17869, Gyeonggi-do, Republic of Korea; futuring.kim@ptu.ac.kr
3. Department of Environmental Science and Ecological Engineering, Korea University, Seoul 02841, Republic of Korea; yson@korea.ac.kr (Y.S.); eepps_korea@korea.ac.kr (S.-W.J.); whijin@korea.ac.kr (W.K.)
4. Department of Forestry, Environment and Systems, Kookmin University, 77 Jeongneung-ro, Seongbuk-gu, Seoul 02707, Republic of Korea; kyeonghlee@kookmin.ac.kr
* Correspondence: leewk@korea.ac.kr; Tel.: +82-3290-3016

**Abstract:** In South Korea, Agriculture, Forestry, and Other Land Use (AFOLU) collates greenhouse gas (GHG) inventories. However, the settlement category lacks a clear definition of land use and activity data. This study proposed a method for examining the settlement spatial extent and constructing activity data to estimate GHG emissions and absorption as a pilot calculation, as well as to provide data for land use classification. Utilizing cadastral maps (CDMs), settlement spatial extents were determined, with settlements occupying approximately 11% of the total land area in 2019, or 9% excluding overlaps. Activity data for settlements were established through a sampling method and analysis of aerial orthoimages from 2000 and 2019. After removing overlaps with digital forest type maps and smart farm maps, settlement activity data covered approximately 18.47% based on CDMs, or 12.66% excluding overlaps. In 2019, $CO_2$ emissions and absorptions were estimated at 622.16 $ktCO_2yr^{-1}$ based on CDMs and 242.16 $ktCO_2yr^{-1}$, excluding overlaps. To enhance GHG inventory calculation consistency and compliance with TACCC principles, clear spatial extents for settlements must be established. This entails constructing activity data and assessing GHG inventories accordingly. GHG inventory statistics should also inform future nationally determined contributions.

**Keywords:** greenhouse gas inventory; AFOLU; settlements spatial extent; activity data; land use; land use change

## 1. Introduction

Since the Paris Agreement's (PA) enforcement in 2021, ratifying countries have been advancing toward limiting global warming to below 2 °C, with ambitions for a 1.5 °C cap, by developing strategies aligned with their Nationally Determined Contributions (NDCs). The "net zero" concept, rooted in Article 4 of the agreement, targets zero net emissions by balancing greenhouse gas (GHG) reductions and absorptions. The 2050 Net Zero Declaration, which has been endorsed by major global players, has fueled debates on achieving net zero emissions, necessitating a systematic approach that includes emissions reduction, absorption, removal, and GHG management via data collection and inventorying [1].

The GHG inventory, following Intergovernmental Panel on Climate Change (IPCC) guidelines, calculates national-level emissions or absorptions from human activities for reporting to the United Nations Framework Convention on Climate Change (UNFCCC). It divides GHG inventories into four sectors, including energy, industrial processes and product use (IPPU), agriculture, forestry, and other land use (AFOLU), and waste, each

with specific calculation methods. Within the AFOLU sector, carbon sinks, which are crucial for reaching 2050 net zero targets, are a focus of research [2]. The current annex countries recognize the significant impact of AFOLU on achieving NDCs and are making efforts to accurately assess absorption and emissions [2–6]. The AFOLU sector tracks land use changes from human activities, calculating GHG inventories for land areas under use and management. It assesses changes in GHG sources and sinks, including biomass, soil, and decaying organic materials, by evaluating land use changes and their impacts. Land is categorized into forest, cropland, grassland, wetland, settlement, and other lands, each with distinct calculation methods for GHGs inventory [7].

The GHG inventory calculations must adhere to each country's monitoring–reporting–verification (MRV) system, aligning with the Paris Agreement's Article 13 transparency system and transparency, accuracy, completeness, comparability, and consistency (TACCC) principles. Article 14 mandates the use of the 2006 IPCC Guidelines (GL) under the Enhanced Transparency Framework (ETF). Inventory assessments should detail sources, assumptions, and methods to allow result reproduction and evaluation [8]. Chapter 2 (National Inventory Report) of the Annex specifies inventory preparation requirements, including national arrangements, the use of IPCC 2006 GL (methodology, parameters, data, analysis, time series, uncertainty, completeness, quality assurance), measurement units (e.g., $tCO_2eq$), report information, and sectoral and GHG time series data [9]. As such, the international community emphasizes the importance of reliable national inventory calculations for enhanced reporting of GHG reduction. In addition to ratifying the UNFCCC (1993), Kyoto Protocol (2002), and PA (2016), South Korea has also submitted its NDCs, which call for a 37% decrease in GHG emissions by 2030 relative to the currently estimated 851 million tons. South Korea established the Greenhouse Gas Inventory and Research Centre (GIR), under the Ministry of Environment, to evaluate the GHG inventory and reduction performance of the NDCs. Biennial update reports (BUR) are submitted with relevant departments for GHG inventory calculation. From 2024, the Paris Agreement parties must submit national inventory reports (NIR) and biennial transparency reports (BTR) to the UNFCCC, highlighting the need for accurate GHG inventory calculations. In addition, the 2023 "Enforcement Decree of The Framework Act on Carbon Neutrality and Green Growth for Coping with Climate Crisis" mandates South Korea to craft a "Basic Plan for Carbon-Neutral Green Growth" nationally and regionally and set and meet GHG reduction targets. Thus, identifying carbon sinks and sources and assessing the GHG inventory at both national and regional levels are critical.

In South Korea's AFOLU sector, four supervisory and six calculating agencies work on GHG inventory estimations using activity data defined for each land category. Yet, settlement categorization faces challenges due to undefined spatial extents and missing activity data, complicating land use and change assessments. The diversity in sizes and types of settlements and carbon sinks within them makes compiling activity data challenging. Austria reports its GHGs inventories for settlements using sample activity data, while Japan calculates and reports using data for specific types. However, settlements are directly affected by human activities, and the amount of land converted to settlements is increasing worldwide. Therefore, it is important to collect activity data and calculate GHG inventory for settlements.

Moreover, in South Korea's AFOLU sector, GHG inventories are calculated using Approach 1, which relies mainly on the area of the land due to the inability to provide explanations for land use changes. Explanations about land use and changes in land use can be conducted at levels 1 to 3 in Approach 1. Approach 2 uses area information of land use changes, whereas Approach 3 describes land use changes spatiotemporally. To report an inventory based on ETF, a GHG inventory calculation at the level of Approach 3 is required. Therefore, in South Korea, to ensure coherence, efforts must be made to reduce disparities resulting from land use classification criteria and data collection and utilization. Various studies in South Korea have attempted to achieve the level of estimation as that in Approach 3 [10–17]. However, inconsistent views among the departments in the govern-

ment have led to an inadequate establishment of consistent criteria for land use classification. These differing views stem from the fact that each department has established and used spatial data with different land classification criteria. The existence of overlapping or missing land parcels has hindered its effective utilization. Therefore, criteria for the utilization and priority of each spatial data should be established to achieve the level of Approach 3. The United States NIR has a priority land use category standard to address the duplication of spatial data and facilitate the utilization of various data sources for consistent criteria [18]. Therefore, South Korea requires uniform criteria for the use of spatial data in land use classification and the establishment of consistent classification standards, which are crucial for the calculation of GHG inventory in the settlements category. Since the 1950s, the area of South Korean settlements has been expanding globally. Therefore, it is important to have accurate land use and GHG calculations [19]. This increase is important in terms of changes in land use and carbon cycling [20–22]. Consequently, there will be an increasing demand for assessing human activities within settlements and the corresponding GHG emission and absorption.

The objectives of this study are as follows: (1) to estimate the area of settlements based on the national MRV system and land use changes, (2) to propose a methodology for constructing activity data for the GHG inventory of settlements category, and (3) to develop foundational data to address potential double-counting issues when estimating the GHG inventory based on constructed activity data. Through this research, the ultimate goal is to provide methodologies for countries like South Korea that require settlements activity data and to further discuss land use classification methods in the AFOLU sector.

## 2. Materials and Methods

In this study, the spatial extent of settlements was determined using cadastral maps (CDMs) from 2000 and 2019. In addition, considering the spatial qualitative synthesis for the national land use classification, spatial data (digital forest type maps [DFTMs], smart farm maps [SFMs]) used in other areas (forests, croplands) were employed to determine the area of activity for settlements and overlapping areas. For the construction of activity data, orthoimages from South Korea, specifically from both 2000 and 2019, were used (Table 1).

**Table 1.** Spatial data for defining settlements, constructing activity data, and analyzing overlap area with other land use categories.

| Spatial Data | Description | Time Series Coverage | Data Type | Reference |
|---|---|---|---|---|
| Cadastral Maps | A detailed map of the cadastral status of the South Korea | 1970s–present (renewed monthly) | Vector | National Spatial Infrastructure Portal (https://www.vworld.kr/ (accessed on 5 March 2021)) |
| Digital Forest type Maps (1:25,000) | A spatialized forest map of the distribution of forests on the land cover | 1st (1971–1974), 2nd (1978–1980), 3rd (1986–1992), 4th (1996–2005), 5th (2006–2010) | Vector | Forest Geospatial Information System (https://fgis.forest.go.kr/ (accessed on 1 April 2023)) |
| Digital Forest type Maps (1:5000) | Large-scale maps of the spatial distribution of forests on the land cover | 2009–2013 (renewed annually) | Vector | |
| Forest aerial photographs | Aerial photographs of South Korea's entire national territory collected in four different periods | 1st (1971–1974), 2nd (1978–1980), 3rd (1986–1992), 4th (1996–2005) | Raster (0.8 m) | Forest Big Data Exchange Platform (https://www.bigdata-forest.kr (accessed on 15 March 2021)) |

**Table 1.** *Cont.*

| Spatial Data | Description | Time Series Coverage | Data Type | Reference |
|---|---|---|---|---|
| Smart Farm Maps | Provides area and attribute information for cropland on the land cover | 2014–2018 | Vector | Agricultural and Rural Affairs Farm map Service (https://agis.epis.or.kr/ (accessed on 1 April 2023)) |
| Orthoimages | Images that have been orthorectified from aerial photographs | 2002–present (Renewed every 2 years) | Raster (urban 12 cm, others 25 cm) | National Geographic Information Institute (https://map.ngii.go.kr/ms/map/NlipMap.do/ (accessed on 25 June 2021)) |

*2.1. Study Area*

South Korea is geographically located in East Asia (between 33.8° N to 39° N latitude and 124.5° E to 130° E longitude), with a total area of 100,401 km² in 2020. The land is categorized into 63,636 km² of forest land, 19,355 km² of cropland, 561 km² of grassland, 6061 km² of wetlands, and 10,789 km² of settlement areas [23]. In 2000, the land use consisted of 65,139 km² of forests, 21,044 km² of cropland, 552 km² of grassland, 5640 km² of wetlands, and 7086 km² of settlement areas [24], leading to South Korea becoming one of the countries with large changes in land use. Since the 1960s, South Korea has experienced rapid industrialization and urbanization due to economic development, leading to decreased forest and cropland and increased settlement areas. This change in land use has led to a concentration in urban areas like Seoul, where 91.9% of South Korea's total population is concentrated [19]. Settlements in South Korea include residential areas, industrial sites, commercial zones, and green spaces, each with distinct types and regional characteristics affecting biomass distribution.

We chose sample areas to curate activity data for the settlements. To determine the sample areas for activity data construction, the entire territory of South Korea was first divided into 393,898 grids, each measuring 500 × 500 m² (Figure 1a). Next, considering the requirements for the sample size design proposed by the IPCC (confidence level of 95%, sampling error of 0.5%), 10% of the total grids comprising 19 land categories in CDMs were selected, resulting in 32,071 sample areas. The sample areas were determined using a systematic sampling method, with targeting grids containing the 19 land categories (Figure 1b,c).

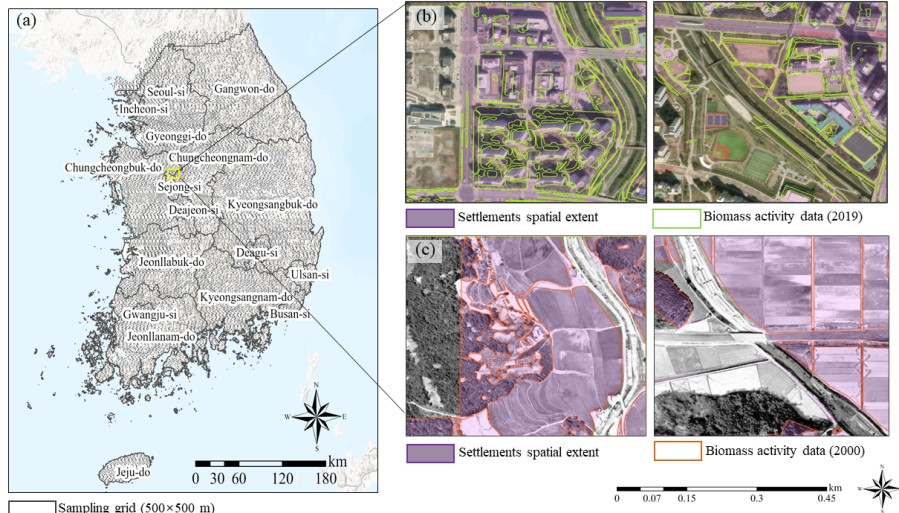

**Figure 1.** Study area and spatial extent for constructing activity data. (**a**) Distribution of sample region for constructing activity data in South Korea; (**b**) current settlements' spatial extent and activity data for biomass; (**c**) the current spatial extent of settlements and land use change is determined by constructing and analyzing activity data from 20 years ago.

### 2.2. Workflow Overview

The workflow of this study comprises the following three steps: (1) Defining the spatial extent of settlements: the IPCC and South Korea's MRV guidelines were reviewed, and available spatial data were selected accordingly. The spatial extent of other land categories' definitions and spatial data was also used to construct a land use matrix. (2) Constructing activity data based on land use change: activity data based on land use change from 2000 to 2019 was constructed by distinguishing between "Land converted to Settlements" (LS) and "Settlements remaining Settlements" (SS). In addition, overlay analysis was employed to identify the areas where activity data spatially overlap, and the areas where activity data do not overlap with other spatial data, such as forest and cropland. (3) The GHG emission and absorption amount for LS and SS were calculated using the GHG inventory calculation method presented by the IPCC. In addition, when calculating the GHG inventory, the inventory, excluding the areas of spatial overlap based on the definition of other land use categories, was calculated, which provides a pilot calculation of the inventory according to the spatial extent setting of settlements (Figure 2).

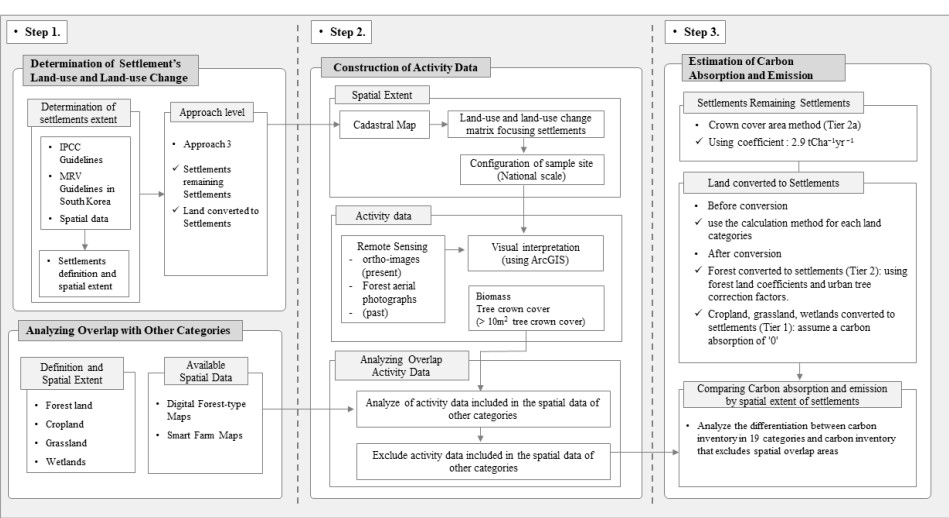

**Figure 2.** Flowchart of this study.

### 2.3. Definition and Spatial Extent of Settlements

Constructing activity data requires clear definitions and spatial extents for land use classification, but South Korea has no precise definitions and spatial extents for settlements [25–27]. Thus, to define the spatial extent of settlements, we examined the 2006 IPCC GL, available spatial data in South Korea and MRV guidelines for land use, land use change, and forestry (LULUCF) sectors, which informed the construction of activity data. In the 2003 GPG-LULUCF and 2006 IPCC GL, "settlements" are defined as "developed lands that include transportation facilities and human habitats, not included in other land use categories". The term also includes trees, i.e., herbaceous plants, such as grass or garden trees, perennial plants, and arboreal plants, seen in rural areas and home gardens [7,28]. The definition of settlements according to the IPCC GL should be adapted to each country's specific context, considering that GHG inventory is calculated according to national standards. In the case of South Korea, this involves applying land use classifications from the Spatial Information Management Law and MRV guidelines from the AFOLU sector [29]. Settlements can be defined as "19 cadastral categories" that do not fall under other land use categories (mineral spring site, salt flat, site, factory site, school site, parking zone, gas station, storage site, road, railroad, embankment, waterways, park, sport site, amusement park, religion site, historic site, grave, and miscellaneous land).

According to the MRV guidelines for the AFOLU sector, land use for grasslands, wetlands, and settlements is defined based on the categories in cadastral statistics. However, definitions from a different source were applied to cropland and forest land. Forest land is

classified according to the definition used in the Forest Basic Statistics (forests with a canopy cover of ≥30% or stands with grown coniferous trees of 1200 per ha and deciduous trees of 1600 per ha) [30], and DFTMs are constructed using related spatial data. In cropland, the range is determined using subcategories, such as paddy fields, fields, and orchards, as used in the Agricultural Area Survey [31], and thematic maps of SFMs have been constructed. Challenges have been experienced in land use classification and GHG inventory estimation due to discrepancies between the definitions of forest and cropland in cadastral statistics, CDMs, basic forest statistics and agricultural land surveys. Therefore, this study mainly used the CDMs to define the spatial extent of settlements and to construct a land use change matrix. This study also included an estimation of carbon absorption and emissions and activity data for settlements, excluding areas overlapping with spatial data (DFTMs, SFMs).

### 2.4. Construction of Land Use and Land Use Change Matrix and Activity Data

We used the subcategories of CDMs to divide the area of LS and SS to construct activity data based on 20 years of land use change. The CDMs subcategories were classified based on land use categories (Table 2). "Other land converted to settlements" was excluded in this study because the spatial extent of settlements in South Korea does not distinguish between settlements and other lands.

**Table 2.** Determination of the spatial extent of settlements using cadastral maps and land use classification of other land use categories.

| Land Use Categories | Categories in Cadastral Map |
|---|---|
| Forest land | Forest land |
| Cropland | Field, paddy, orchard |
| Grassland | Pasture |
| Wetlands | River, ditch, reservoir, fish farm |
| Settlements | Mineral spring site, salt flat, site, factory site, school site, parking zone, gas station, storage site, road, railroad, embankment, waterways, park, Sport site, amusement park, religion site, historic site, grave, miscellaneous land |

We focused on constructing activity data for biomass to address the limited information on the location and characteristics of carbon sinks within settlements in South Korea, as well as the absence of developed GHG emission and absorption coefficients. This approach followed the 2006 IPCC GL that provide default coefficients for biomass. Therefore, biomass activity data was constructed for tree crown-cover area. Activity data for tree crown-cover within settlements, considering the small-scale and dispersed characteristics, were constructed using high-spatial resolution ortho aerial imagery (25 cm) and visual interpretation by the ArcGIS program. The activity data for visual interpretation were set for the crown-cover area with an area of ≥10 m$^2$. Ortho aerial photographs provided by the National Geographic Information Institute were used for 2019 data. However, because ortho aerial photographs have only been available since 2010, the Fourth National Forest Aerial Photography (1996–2005) was used for the 2000 data.

### 2.5. Comparison of Activity Data Considering Other Land Use Categories

The IPCC and Article 4.13 of the Paris Agreement emphasize the importance of preventing double-counting in GHGs inventory and ensuring TACCC. In South Korea, differing land use classification definitions cause overlaps, challenging TACCC adherence. To mitigate this, we calculated settlement biomass activity data areas, excluding overlaps with DFTMs or SFMs.

### 2.6. Estimation of Carbon Emission and Absorption

The IPCC GL outlines three levels for estimating $CO_2$ emissions and absorption, including Tier 1, Tier 2, and Tier 3. Evaluation complexity and accuracy enhance from Tier 1 through Tier 3, facilitating more precise assessments. The choice of tier level hinges on the availability and accessibility of data regarding biomass, dead organic matter, and soil [7]. For SS, Tier 1 presumes constant carbon absorption, while Tier 2 assesses absorption based on changes in tree crown-cover or individual tree growth, using default coefficients from the 2006 IPCC GL for carbon absorption. Thus, this study adopted Tier 2a for SS, allowing carbon absorption evaluation at a default coefficient of 2.9 tC·ha$^{-1}$·yr$^{-1}$ for each ha of tree crown-cover area (Equation (1)).

$$\Delta C_G = \sum_{i,j} AT_{i,j} \cdot CRW_{i,j} \tag{1}$$

where $\Delta C_G$ = annual carbon accumulation attribution to biomass increment in SS (tC·yr$^{-1}$); $AT_{i,j}$ = total crown-cover area of class $i$ woody perennial type $j$ (ha); and $CRW_{i,j}$ = crown-cover area-based growth rate of class $i$ in woody perennial type $j$ (tC·ha$^{-1}$·yr$^{-1}$).

For LS, the tier level was determined by considering whether nationally specific emission and absorption coefficients have been developed for each land use category. For forest land converted to settlements (FS), the carbon stock in forest land before conversion was estimated using Equation (2) and nationally specific emission and absorption coefficients of Tier 2. Supplementary data on the area (ha) of national-level forest type and stock volume were obtained from the forest statistics [32].

$$C = \sum_{i,j} \left\{ A_{i,j} \times V_{i,j} \times BCEF_{i,j} \times (1 + R_{i,j}) \times CF_{i,j} \right\} \tag{2}$$

where $C$: Carbon stock, $A_{i,j}$: Forest land area, $V_{i,j}$: Growing stock, $BCEF_{i,j}$: Biomass expansion factor, $R_{i,j}$: Root-shoot ratio, $CF_{i,j}$: Carbon fraction, $i$: Forest type, and $j$:Climate zone.

For cropland converted to settlements (CS) and grassland converted to settlements (GS), carbon stock before the conversion were estimated using Tier 1 and default factors suggested in the 2006 IPCC GL (Table 3). Wetlands converted to settlements (WS) were excluded from this study, considering the lack of biomass calculating method in the IPCC GL. The Tier 1 method to estimate the amount of land absorbing GHG prior to conversion to CS and GS includes multiplying the basic emission absorption coefficient with the area of the respective land.

**Table 3.** Greenhouse gas emission absorption coefficient used to calculate the inventory for settlements converted from other land use categories.

| Division | Supplementary Data, Emission Absorption Factors | | Reference |
|---|---|---|---|
| Forest land converted to settlements | Area and stock (ha, m$^3$) | Conifer, deciduous, mixed forest | [32] |
| | Basic wood density (t d.m. m$^{-3}$) | Conifer: 0.46, deciduous: 0.68, mixed: 0.57 | [33] |
| | Biomass expansion factor | Conifer: 1.43, deciduous: 1.51, mixed: 1.47 | |
| | Root–shoot ratio | Conifer: 0.27, deciduous: 0.36, mixed: 0.32 | |
| | Carbon fraction | 0.5 | |
| Croplands converted to settlements | 2006 IPCC default coefficient: 4.7 C ha$^{-1}$) | | [7] |
| Grassland converted to settlements | 2006 IPCC default coefficient: 13.5 t.d.m.ha$^{-1}$ for warm temperate-wet climate zone, non-woody biomassCarbon fraction: 0.5 tC (tonne d.m.)$^{-1}$ | | |

Furthermore, we evaluated the carbon emission and absorption, excluding areas with spatial overlaps between other land use categories and settlements. This analy-

sis verified the carbon inventory for settlements according to the spatial division of the land use category.

## 3. Results

### 3.1. Construction of Land Use and Land Use Change Matrix

In 2019, the settlement area was 108.17 million ha, accounting for 10.74% of South Korea's total land area. By region, the settlement area of Gyeonggi-do was the largest at 200,180 ha, whereas Sejong-si was the smallest at 7599 ha. The region with the highest settlement ratio was Seoul-si (63.9%), whereas that with the lowest was Gangwon-do (4.6%). The sampling ratio for constructing settlement activity data by province ranged from 9.5% to 11.7%, which was used to estimate the total activity data area by province. Between 2000 and 2019, there was a land use change resulted in 734,300 ha (68% of current settlements) of SS and 347,300 ha (32%) of LS, at an annual conversion of 17,365 ha from other land categories (Table 4).

**Table 4.** The area and ratio of settlements by region in South Korea and the sampling ratio.

| Division | Total Land Area (ha) | Area of Land Converted to Settlements (ha) | Area of Settlements Remaining Settlements (ha) | Total Area of Settlements (ha) | Settlements Ratio (%) | Sampling Ratio (%) |
|---|---|---|---|---|---|---|
| Gangwon-do | 1,682,968 | 28,274 | 49,799 | 78,073 | 4.6 | 10.4 |
| Gyeonggi-do | 1,019,527 | 77,346 | 123,917 | 201,263 | 19.7 | 10.4 |
| Gyeongsangnam-do | 1,054,055 | 30,371 | 72,662 | 103,033 | 9.8 | 10.9 |
| Gyeonsangbuk-do | 1,903,403 | 47,170 | 71,313 | 118,482 | 6.2 | 10.1 |
| Gwangju-si | 50,113 | 4701 | 11,232 | 15,932 | 31.8 | 9.5 |
| Daegu-si | 88,349 | 6923 | 16,456 | 23,378 | 26.5 | 10.2 |
| Daejeon-si | 53,966 | 3574 | 11,722 | 15,296 | 28.3 | 10.0 |
| Busan-si | 77,007 | 7788 | 19,961 | 27,749 | 36.0 | 11.1 |
| Seoul-si | 60,523 | 2997 | 35,662 | 38,659 | 63.9 | 10.1 |
| Sejong-si | 46,491 | 4442 | 3157 | 7599 | 16.3 | 9.6 |
| Ulsan-si | 106,209 | 5830 | 13,745 | 19,575 | 18.4 | 9.9 |
| Incheon-si | 106,523 | 9233 | 28,127 | 37,360 | 35.1 | 11.7 |
| Jeollanam-do | 1,234,809 | 36,902 | 84,582 | 121,483 | 9.8 | 11.4 |
| Jeollabuk-do | 806,984 | 14,171 | 66,270 | 80,441 | 10.0 | 10.1 |
| Jeju-si | 185,021 | 10,973 | 16,335 | 27,308 | 14.8 | 10.2 |
| Chungcheongnam-do | 824,617 | 33,426 | 64,569 | 97,995 | 11.9 | 10.4 |
| Chungcheongbuk-do | 740,695 | 23,212 | 44,814 | 68,027 | 9.2 | 10.1 |
| Total | 10,041,260 | 347,331 | 734,322 | 1,081,653 | 10.8 | 10.5 |

The ratio of LS by province was as follows: Sejong-si had the highest proportion at the regional level with 58.46%, followed by Jeju-si (40.18%) and Gyeonggi-do (38.43%). Seoul-si had the highest proportion of SS at 92.25%, followed by Jeollanam-do (82.38%), Daejeon-si (76.67%), Incheon-si (75.29%), Busan-si (71.93%), and Daegu-si (70.39%). This indicates that in metropolitan cities where land development occurred as of the year 2000, a high proportion of SS was observed. In areas where land development has been more recent, a higher proportion of LS was evident (Figure 3).

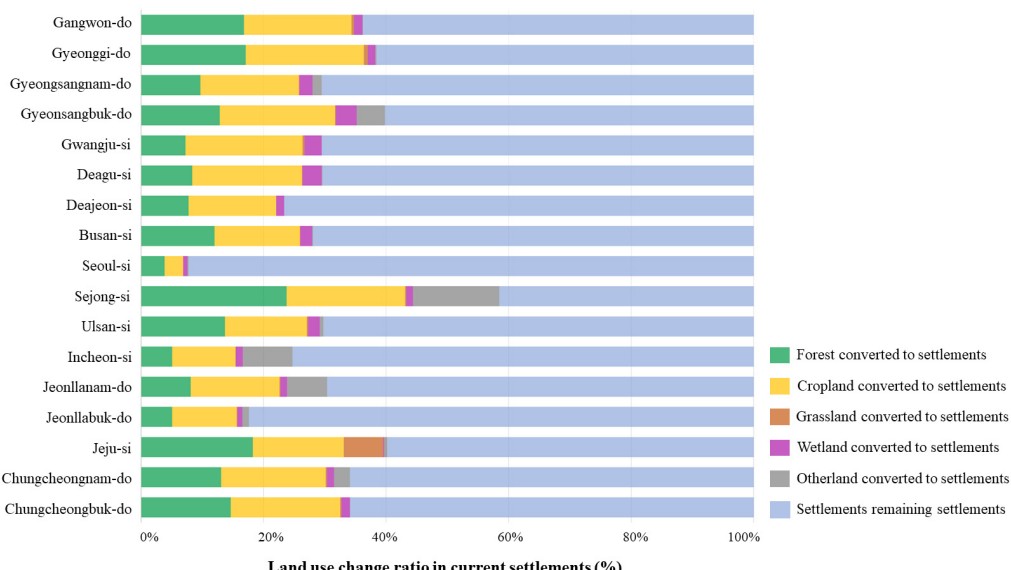

**Figure 3.** Ratio of land use and land use change in settlements according to cadastral maps.

### *3.2. Comparison of Activity Data Considering Other Land Use Categories*

#### 3.2.1. Land Converted to Settlements

We calculated the activity data area for settlements at the national level by multiplying the area of activity data constructed, based on land use change categories, with the sample extraction ratio for each region in Section 3.1 (Table 4). In the case of LS, before conversion to settlements (in 2000), the total biomass cover of other land categories was 0.23 million ha (FS 0.09 million ha, CS 0.10 million ha, and GS 0.04 million ha), and the crown-cover area of settlements after conversion was observed to be 0.07 million ha (in 2019). The spatial overlap between LS conversion activity data and both DFTMs and SFMs was 0.02 million ha. This accounted for 9.9% of total activity data for LS before conversion. After the conversion to settlements, there was an overlap of 0.04 million ha in the activity data, representing 52.1% of the total activity data. Gyeonggi-do had the largest overlap on a regional level (0.50 million ha before conversion, 0.06 million ha after conversion), whereas Gwangju-si had the smallest overlap (259 ha before conversion, 412 ha after conversion) (Table 5).

**Table 5.** Construction of activity data within land converted to settlements and results derived from the area of activity data, excluding overlap with DFTMs and SFMs.

| | | | | | | Unit: ha |
|---|---|---|---|---|---|---|
| | **Before (2000)** | | | **After (2019)** | | |
| **Division** | **19 Categories in CDM** | **Overlapping** | **Exclude Overlapping** | **19 Categories in CDM** | **Overlapping** | **Exclude Overlapping** |
| Gangwon-do | 19,226 | 3362 | 15,864 | 7494 | 4023 | 3471 |
| Gyeonggi-do | 52,571 | 6245 | 46,325 | 13,221 | 6710 | 6511 |
| Gyeongsangnam-do | 14,922 | 1099 | 13,823 | 5603 | 1793 | 3810 |
| Gyeonsangbuk-do | 28,526 | 2549 | 25,977 | 9046 | 3978 | 5067 |
| Gwangju-si | 3997 | 259 | 3738 | 1165 | 412 | 753 |
| Daegu-si | 4720 | 505 | 4214 | 1509 | 807 | 702 |
| Daejeon-si | 2620 | 381 | 2239 | 1067 | 631 | 437 |
| Busan-si | 5774 | 273 | 5501 | 1278 | 664 | 614 |
| Seoul-si | 1452 | 506 | 946 | 1335 | 820 | 515 |

**Table 5.** *Cont.*

| | Before (2000) | | | After (2019) | | Unit: ha |
|---|---|---|---|---|---|---|
| Division | 19 Categories in CDM | Overlapping | Exclude Overlapping | 19 Categories in CDM | Overlapping | Exclude Overlapping |
| Sejong-si | 3093 | 298 | 2795 | 1252 | 952 | 300 |
| Ulsan-si | 5082 | 162 | 4920 | 841 | 180 | 662 |
| Incheon-si | 3310 | 487 | 2822 | 2328 | 965 | 1363 |
| Jeollanam-do | 26,499 | 1959 | 24,540 | 7549 | 3020 | 4528 |
| Jeollabuk-do | 10,515 | 726 | 9789 | 1536 | 615 | 921 |
| Jeju-si | 9067 | 1393 | 7674 | 2496 | 1391 | 1105 |
| Chungcheongnam-do | 23,356 | 1768 | 21,588 | 4826 | 2682 | 2143 |
| Chungcheongbuk-do | 16,481 | 1005 | 15,476 | 4642 | 2521 | 2121 |
| Total | 231,209 | 22,977 | 208,232 | 67,187 | 32,164 | 35,022 |

DFTMs: digital forest type maps; SFMs: smart farm maps.

### 3.2.2. Settlements Remaining Settlements

The crown-cover area of SS, which was 0.1 million ha in 2000, increased by 0.03 million ha to 0.13 million ha in 2019. In the regional level, Gyeonggi-do showed the largest increase in crown-cover area, from 16.8 thousand ha in 2000 to 18.2 thousand ha in 2019. The region with the most significant increase in crown-cover area was Gyeongsangnam-do, with an increase of 7.3 thousand ha. In contrast, Jeollabuk-do showed a decrease of 695 ha to 7.2 thousand ha in 2019 compared to the year 2000. The spatial overlap in activity data between SS and both DFTMs and SFMs was 0.05 million ha in 2000 (52.7% of the total past activity data for SS) and 0.04 million ha in 2019 (28.3% of the current activity data for SS). At the regional level, Gyeongsangbuk-do had the highest spatial overlap, with 0.08 million ha in 2000 and 0.06 million ha in 2019, while Sejong-si showed the lowest overlap, increasing from 360 ha in 2000 to 770 ha in 2019 (Table 6).

**Table 6.** Construction of activity data within settlements remaining settlements and results derived from the area of activity data, excluding overlap with DFTMs and SFMs.

| | Before (2000) | | | After (2019) | | Unit: ha |
|---|---|---|---|---|---|---|
| Division | 19 Categories in CDM | Overlapping | Exclude Overlapping | 19 Categories in CDM | Overlapping | Exclude Overlapping |
| Gangwon-do | 13,185 | 7152 | 7286 | 13,582 | 5899 | 6429 |
| Gyeonggi-do | 16,832 | 4410 | 11,657 | 18,195 | 5175 | 13,785 |
| Gyeongsangnam-do | 9886 | 7202 | 5536 | 17,193 | 4350 | 9991 |
| Gyeonsangbuk-do | 12,726 | 7957 | 6830 | 15,979 | 5896 | 8021 |
| Gwangju-si | 1126 | 674 | 858 | 2188 | 268 | 1514 |
| Daegu-si | 1849 | 1655 | 1067 | 3607 | 782 | 1952 |
| Daejeon-si | 2085 | 1197 | 1211 | 2664 | 874 | 1467 |
| Busan-si | 2688 | 1636 | 2052 | 3638 | 636 | 2002 |
| Seoul-si | 2690 | 1941 | 1988 | 7248 | 702 | 5307 |
| Sejong-si | 185 | 360 | 107 | 823 | 77 | 463 |
| Ulsan-si | 747 | 472 | 483 | 1874 | 264 | 1402 |

| | | | | | Unit: ha |
|---|---|---|---|---|---|
| **Division** | **Before (2000)** | | | **After (2019)** | | |
| | **19 Categories in CDM** | **Overlapping** | **Exclude Overlapping** | **19 Categories in CDM** | **Overlapping** | **Exclude Overlapping** |
| Incheon-si | 1098 | 927 | 983 | 4338 | 115 | 3411 |
| Jeollanam-do | 10,536 | 4905 | 6535 | 14,346 | 4001 | 9441 |
| Jeollabuk-do | 7909 | 2936 | 5269 | 7214 | 2640 | 4278 |
| Jeju-si | 2550 | 1425 | 1442 | 3488 | 1108 | 2063 |
| Chungcheongnam-do | 7102 | 4512 | 4742 | 8606 | 2360 | 4093 |
| Chungcheongbuk-do | 8017 | 3984 | 5437 | 8488 | 2580 | 4504 |
| Total | 101,209 | 53,346 | 63,482 | 133,469 | 37,727 | 80,123 |

DFTMs: digital forest type maps; SFMs: smart farm maps.

### 3.2.3. Biomass Change Ratio by Land Use and Land Use Change

In the case of LS, the national level of before-conversion biomass ratio to the current settlements area was found to be 21.4%, with variations observed across regions. Notably, Sejong-si and Jeju-do had the highest biomass ratios at 40.7% and 33.2%, respectively, while Seoul-si had the lowest ratio at 3.8% (Figure 4a). After excluding spatial overlap areas, the before-conversion biomass ratio of LS was found to be 19.3%. This indicates that the inclusion or exclusion of overlapping areas can lead to significant differences in the area covered by activity data (Figure 4d). Jeju-do had the largest difference in biomass cover area (5.1%), while Jeollabuk-do and Ulsan-si had the smallest differences, both at 0.9%.

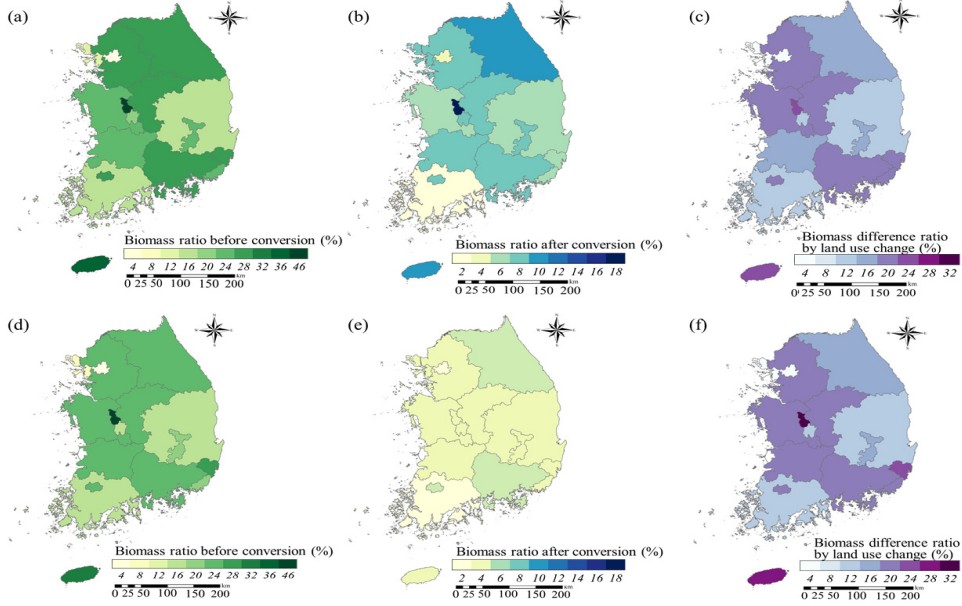

**Figure 4.** Biomass ratio before and after LS conversion. (**a**) Biomass ratio of the converted land before CDMs, (**b**) biomass ratio of the settlement after CDMs, and (**c**) biomass difference ratio before and after conversion. (**d**) Biomass ratio of converted land excluding the overlapping areas of DFTMs and SFMs. (**e**) Biomass ratio of settlements after conversion, excluding the overlapping areas of DFTMs and SFMs. (**f**) Biomass difference ratio before and after conversion.

When converted to settlements, the national level of crown-cover ratio decreased by 15.2% (Figure 4c). Regionally, Sejong-si showed the highest decrease at 16.5%, and Jeollabuk-do the lowest at 1.9% (Figure 4b). After conversion to settlements and the

excluding of spatial overlap areas, the national level of biomass ratio was found to be 3.2%, indicating a decrease of about half (Figure 4e). The crown-cover ratio, with overlaps excluded, was highest in Gwangju-si at 4.7% and lowest in Jeollabuk-do at 1.1%. The exclude of overlapping areas ratio resulted in the most significant difference in crown-cover area ratio in Sejong-si (Figure 4f).

In the case of SS, the national level of crown-cover ratio increased from 9.4% in 2000 (Figure 5a) to 12.3% in 2019 (Figure 5b). Regionally, Gangwon-do had the highest ratio at both timepoints (16.9%, 17.4%). Seoul-si had the highest growth ratio, increasing from 7.0% to 18.7%, while Jeollabuk-do had a decrease in ratio from 9.8% to 9.0%. After excluding spatial overlap areas, the before ratio was 5.9% (Figure 5d), and the current ratio was 7.4% (Figure 5e), showing differences of 3.5% and 4.9%, respectively. After excluding overlapping areas, the crown-cover ratio of SS decreased in Gangwon-do, Busan, Jeollabuk-do, Chungcheongnam-do, and Chungcheongbuk-do. Therefore, it was also confirmed for SS that there are significant differences in the activity data depending on whether the overlapping areas with DFTMs and SFMs are included or excluded.

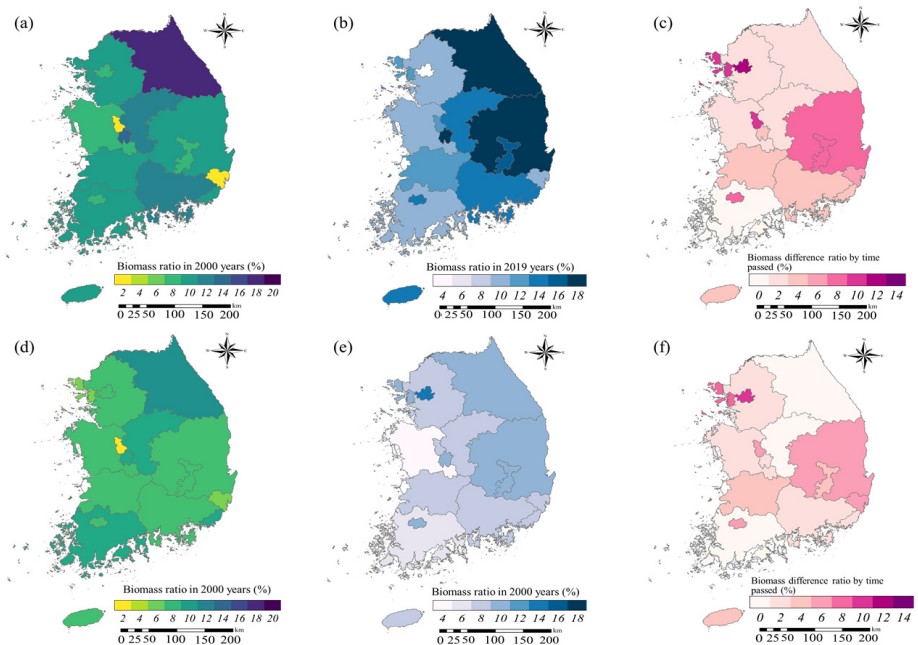

**Figure 5.** Biomass ratio in SS in 2000 and 2019. (**a**) Settlement biomass ratio in 2000 based on CDMs. (**b**) Settlement biomass ratio in 2019 based on CDMs. (**c**) Biomass difference ratio before and after conversion. (**d**) Biomass ratio in 2000, excluding the overlapping areas of DFTMs and SFMs. (**e**) Biomass ratio in 2019, excluding the overlapping areas of DFTMs and SFMs. (**f**) Biomass difference ratio.

### 3.3. Estimation of Carbon Emission and Absorption

3.3.1. Land Converted to Settlements

Calculating $CO_2$ emissions in LS using the Tier 1 method showed that 15,941.22 kt$CO_2$ was emitted over a period of 20 years, with an annual $CO_2$ emission rate of 797.06 kt$CO_2$yr$^{-1}$ (Table 7). This corresponds to 1.82% of South Korea's 2019 $CO_2$ net absorption in the AFOLU sector, which is estimated at 43.9 Mt$CO_2$yr$^{-1}$ [34] and represents 18.12% of the total emission of 4.4 Mt$CO_2$yr$^{-1}$.

By region, $CO_2$ emission in Gyeonggi-do was 4147.40 kt$CO_2$, with an annual emission of 207.37 kt$CO_2$yr$^{-1}$. Furthermore, $CO_2$ emissions varied according to the proportion of land use types before conversion to settlement, with FS making the largest contribution to $CO_2$ emissions. Depending on the regional characteristics, in Jeollanam-do, areas with a small area of forest land and a large area of cropland before conversion had significant emissions due to CS. Therefore, it was crucial to accurately determine the converted land types to calculate $CO_2$ emissions in LS.

**Table 7.** GHG emissions of the settlements converted from other lands, and that after excluding the overlapping areas of other spatial data.

| Division | CDMs | | | | | Without Other Spatial Data | | | | |
|---|---|---|---|---|---|---|---|---|---|---|
| | Forest (ktCO$_2$) | Cropland (ktCO$_2$) | Grassland (ktCO$_2$) | Total CO$_2$ Emission (ktCO$_2$) | Annual CO$_2$ Emission (ktCO$_2$yr$^{-1}$) | Forest (ktCO$_2$) | Cropland (ktCO$_2$) | Grassland (ktCO$_2$) | Total CO$_2$ Emission (ktCO$_2$) | Annual CO$_2$ Emission (ktCO$_2$yr$^{-1}$) |
| Gangwon-do | 1438 | 98 | 78 | 1614 | 81 | 967 | 76 | 65 | 1108 | 55 |
| Gyeonggi-do | 3609 | 375 | 163 | 4147 | 207 | 2660 | 336 | 141 | 3137 | 157 |
| Gyeongsangnam-do | 856 | 108 | 56 | 1019 | 51 | 712 | 98 | 51 | 862 | 43 |
| Gyeonsangbuk-do | 1618 | 215 | 112 | 1945 | 97 | 1260 | 187 | 98 | 1545 | 77 |
| Gwangju-si | 105 | 47 | 12 | 164 | 8 | 72 | 41 | 11 | 125 | 6 |
| Daegu-si | 147 | 46 | 25 | 218 | 11 | 81 | 40 | 21 | 142 | 7 |
| Daejeon-si | 161 | 21 | 8 | 190 | 10 | 102 | 20 | 7 | 130 | 7 |
| Busan-si | 166 | 48 | 44 | 257 | 13 | 130 | 46 | 37 | 212 | 11 |
| Seoul-si | 126 | 9 | 3 | 138 | 7 | 51 | 9 | 2 | 62 | 3 |
| Sejong-si | 245 | 19 | 11 | 274 | 14 | 197 | 18 | 10 | 225 | 11 |
| Ulsan-si | 301 | 35 | 22 | 358 | 18 | 276 | 34 | 21 | 330 | 17 |
| Incheon-si | 183 | 21 | 25 | 229 | 11 | 101 | 18 | 19 | 138 | 7 |
| Jeollanam-do | 881 | 246 | 159 | 1287 | 64 | 566 | 212 | 127 | 905 | 45 |
| Jeollabuk-do | 447 | 108 | 30 | 585 | 29 | 350 | 94 | 28 | 472 | 24 |
| Jeju-si | 668 | 49 | 56 | 773 | 39 | 430 | 40 | 49 | 520 | 26 |
| Chungcheongnam-do | 1186 | 172 | 131 | 1489 | 75 | 926 | 151 | 118 | 1196 | 60 |
| Chungcheongbuk-do | 1077 | 122 | 55 | 1254 | 63 | 932 | 109 | 49 | 1089 | 55 |
| Total | 13,213 | 1738 | 990 | 15,941 | 797 | 9812 | 1530 | 854 | 12,196 | 610 |

Excluding the areas overlapping with DFTMs and SFMs, the CO$_2$ emissions in LS over 20 years was 12,196.43 ktCO$_2$, with an annual CO$_2$ emission of 609.82 ktCO$_2$yr$^{-1}$. When comparing the LS values with the results for 19 cadastral land categories, a difference in emissions was observed over 20 years, with a total emission of 3744.79 ktCO$_2$ and an annual emission of 187.24 ktCO$_2$yr$^{-1}$. By region, Gyeonggi-do showed the largest difference (total emission of 1010.64 ktCO$_2$ and annual emission of 50.53 ktCO$_2$yr$^{-1}$), whereas Gwangju-si showed the smallest difference (total emission of 39 ktCO$_2$ and annual emission of 2 ktCO$_2$yr$^{-1}$). Furthermore, the difference in CO$_2$ emissions per ha for settlements converted from other lands was 0.17 tCO$_2$yr$^{-1}$ ha$^{-1}$. This difference was smaller than that obtained after removing overlapping areas. This is due to the reduction in the area of the total settlement and total area converted from other lands, resulting in a decrease in emissions due to the removal of overlapping areas.

### 3.3.2. Settlements Remaining Settlements

In 2000, the national annual CO$_2$ absorption of SS was 1076.2 ktCO$_2$yr$^{-1}$, which increased to 1419.2 ktCO$_2$yr$^{-1}$ in 2019. Over a period of 20 years, the total annual CO$_2$ absorption change showed an increase of 343.0 ktCO$_2$yr$^{-1}$, with an annual increase of 17.2 ktCO$_2$yr$^{-1}$. This corresponds to 3.2% of South Korea's CO$_2$ net absorption in the AFOLU sector in 2019 (Table 8). By region, Gyeonggi-do showed the largest CO$_2$ absorption with 179.0 ktCO$_2$yr$^{-1}$ in 2000 and 193.5 ktCO$_2$yr$^{-1}$ in 2019, whereas Sejong-si showed the smallest absorption with 2.0 ktCO$_2$yr$^{-1}$ in 2000 and 8.8 ktCO$_2$yr$^{-1}$ in 2019. Furthermore, the total CO$_2$ absorption change over 20 years in settlements maintained as regional settlements increased significantly in Gyeongsangnam-do, reaching 77.70 ktCO$_2$yr$^{-1}$, with an annual increase of 3.88 ktCO$_2$yr$^{-1}$. In contrast, Jeollanam-do showed a total CO$_2$ absorption change of $-7.39$ ktCO$_2$yr$^{-1}$, with an annual decrease of $-0.37$ ktCO$_2$yr$^{-1}$.

The CO$_2$ absorption of SS excluding overlapping areas was 675.0 ktCO$_2$yr$^{-1}$ in 2000, which increased by 8.9 ktCO$_2$yr$^{-1}$ annually to 852.0 ktCO$_2$yr$^{-1}$ in 2019. SS results, excluding areas overlapping with DFTMs and SFMs, showed a difference of 401.2 ktCO$_2$yr$^{-1}$ in

the past and 567.2 $ktCO_2yr^{-1}$ currently. By region, Gyeongsangbuk-do showed the largest difference with 62.7 $ktCO_2yr^{-1}$ in 2000 and 84.6 62.7 $ktCO_2yr^{-1}$ in 2019. Furthermore, the difference in $CO_2$ absorption per ha showed a trend similar to that of LS and was estimated to be 0.5 $tCO_2yr^{-1}ha^{-1}$ lower as a result of excluding overlapping areas.

**Table 8.** GHG emission in settlements remaining settlements and that of settlements, excluding overlapping areas of other spatial data.

| Division | CDMs | | | | Without Other Spatial Data | | | |
|---|---|---|---|---|---|---|---|---|
| | Past $CO_2$ Absorption ($ktCO_2yr^{-1}$) | Present $CO_2$ Absorption ($ktCO_2 yr^{-1}$) | Total change in $CO_2$ Absorption ($\Delta ktCO_2 yr^{-1}$) | Annual Change in $CO_2$ Absorption ($ktCO_2yr^{-1}$) | Past $CO_2$ Absorption ($ktCO_2yr^{-1}$) | Present $CO_2$ Absorption ($ktCO_2 yr^{-1}$) | Total Change in $CO_2$ Absorption ($\Delta ktCO_2 yr^{-1}$) | Annual Change in $CO_2$ Absorption ($ktCO_2yr^{-1}$) |
| Gangwon-do | 140 | 144 | 4 | 0 | 78 | 68 | −9 | −1 |
| Gyeonggi-do | 179 | 194 | 15 | 1 | 124 | 147 | 23 | 1 |
| Gyeongsangnam-do | 105 | 183 | 78 | 4 | 59 | 106 | 47 | 2 |
| Gyeonsangbuk-do | 135 | 170 | 35 | 2 | 73 | 85 | 13 | 1 |
| Gwangju-si | 12 | 23 | 11 | 1 | 9 | 16 | 7 | 0 |
| Daegu-si | 20 | 38 | 19 | 1 | 11 | 21 | 9 | 1 |
| Daejeon-si | 22 | 28 | 6 | 0 | 13 | 16 | 3 | 0 |
| Busan-si | 29 | 39 | 10 | 1 | 22 | 21 | −1 | 0 |
| Seoul-si | 29 | 77 | 49 | 2 | 21 | 56 | 35 | 2 |
| Sejong-si | 2 | 9 | 7 | 0 | 1 | 5 | 4 | 0 |
| Ulsan-si | 8 | 20 | 12 | 1 | 5 | 15 | 10 | 1 |
| Incheon-si | 12 | 46 | 35 | 2 | 11 | 36 | 26 | 1 |
| Jeollanam-do | 112 | 153 | 41 | 2 | 70 | 100 | 31 | 2 |
| Jeollabuk-do | 84 | 77 | −7 | 0 | 56 | 46 | −11 | −1 |
| Jeju-si | 27 | 37 | 10 | 1 | 15 | 22 | 7 | 0 |
| Chungcheongnam-do | 76 | 92 | 16 | 1 | 50 | 44 | −7 | 0 |
| Chungcheongbuk-do | 85 | 90 | 5 | 0 | 58 | 48 | −10 | −1 |
| Total | 1076 | 1419 | 343 | 17 | 675 | 852 | 177 | 9 |

### 3.3.3. Overall $CO_2$ Inventory

We evaluated the $CO_2$ emission and absorption of the entire settlements in South Korea by combining the results of LS and SS. In 2000, the $CO_2$ inventory was 296.3 $ktCO_2yr^{-1}$. In 2019, $CO_2$ absorption was 622.2 $ktCO_2yr^{-1}$. Furthermore, the annual change in $CO_2$ inventory was an increment of 17.2 $ktCO_2yr^{-1}$. This corresponds to 1.6% of the total $CO_2$ absorption of 39.6 $MtCO_2yr^{-1}$ in South Korea's AFOLU sector in 2019. The $CO_2$ inventory absorption relative to the settlement area (ha) in 2019 was 0.58 $tCO_2yr^{-1}$ $ha^{-1}$. By region, Seoul-si showed the highest $CO_2$ absorption rate at 1.82 $tCO_2yr^{-1}$ $ha^{-1}$. In contrast, $CO_2$ was emitted in Gyeonggi-do (13.90 $tCO_2yr^{-1}$ $ha^{-1}$), Jeju-do (0.06 $tCO_2yr^{-1}$ $ha^{-1}$), and Sejong-si (0.65 $tCO_2yr^{-1}$ $ha^{-1}$) (Figure 6a–c).

After removing overlapping areas in the spatial data of DFTMs and SFMs, the $CO_2$ inventory showed an emission of 122.0 $ktCO_2yr^{-1}$ in 2000, whereas an absorption of 242.1 $ktCO_2yr^{-1}$ was observed in 2019. The difference in absorption between 2000 (−418.3 $ktCO_2yr^{-1}$) and 2019 (380.0 $ktCO_2yr^{-1}$) was 798.3 $ktCO_2yr^{-1}$. Thus, the $CO_2$ inventory in 2019 based on the removal of overlapping regions was estimated to account for 0.61% of South Korea's total $CO_2$ absorption from emissions and absorptions in the AFOLU sector. It was confirmed that the evaluation of $CO_2$ inventory in the settlement category may vary depending on the land use classification results for overlapping areas. Furthermore, when considering overlapping areas in the $CO_2$ inventory per settlement area, a difference of 0.4 $tCO_2yr^{-1}ha^{-1}$ was observed in absorption. These differences varied based on the region. Gyeongsangnam-do showed the largest difference (0.7 $tCO_2yr^{-1}ha^{-1}$), followed by Gangwon-do (0.7 $tCO_2yr^{-1}ha^{-1}$) and Daejeon-si (0.6 $tCO_2yr^{-1}ha^{-1}$) (Figure 6d–f).

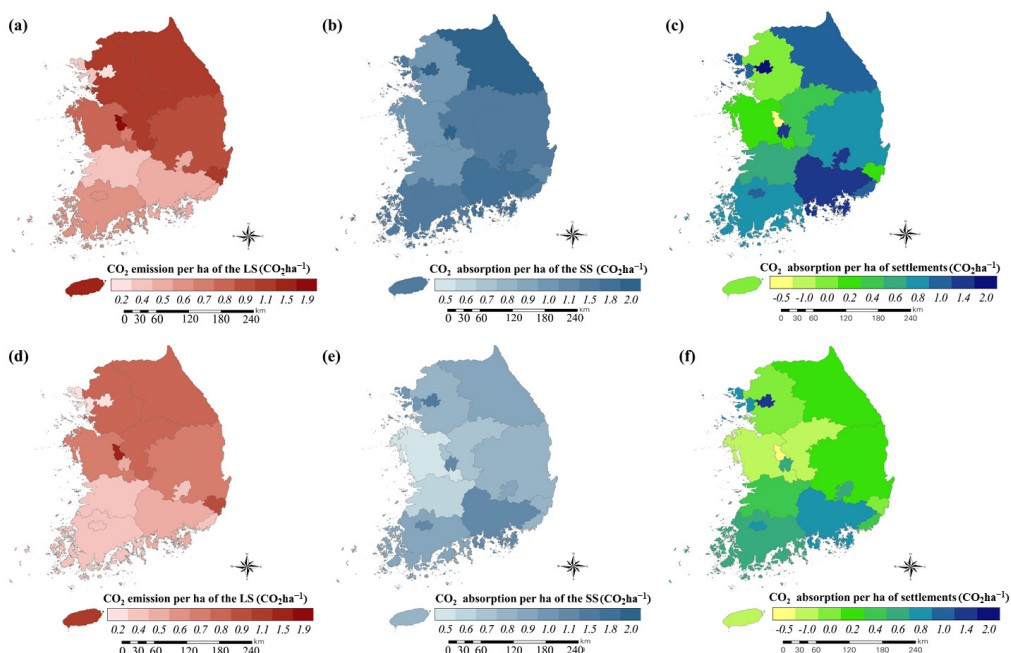

**Figure 6.** GHG emissions and absorption per settlement area according to the spatial extent of the settlement. (**a**) GHG emission per unit area of settlements converted from other lands based on CDMs. (**b**) GHG absorption per unit area of settlements remaining settlements based on CDMs. (**c**) GHG absorption per unit area of settlements that combine GHG emissions and absorption based on CDMs. (**d**) GHG emission of settlements converted from other lands, excluding the overlapping areas of DFTMs and SMFMs. (**e**) GHG absorption of settlements remaining settlements, excluding the overlapping areas of DFTMs and SMFMs. (**f**) GHG absorption per unit area of settlements that combine GHG emissions and absorption, excluding the overlapping areas of DFTMs and SMFMs.

## 4. Discussion

### 4.1. Definition of Settlement and Spatial Extent Setting

Utilizing cadastral information and following South Korea's MRV guidelines for the GHGs inventory of settlements, we defined 19 cadastral land categories not encompassed by other land categories in the CDMs. However, the determined spatial extent for settlements includes overlapping areas with forest land and cropland. The MRV guidelines for the AFOLU sector in South Korea differentiate land cover for forest land and cropland from land use classification for wetlands, grasslands, settlements, and other lands. Therefore, spatial overlaps occur when thematic maps (DFTMs, SFMs) overlap with CDMs that include 19 cadastral land categories [35]. Efforts to resolve the challenges of overlapping and missing land area and activity data, caused by the mixing of land cover and land use concepts, have been attempted in South Korea. Nevertheless, these issues continue to hinder GHG inventory assessments [36]. "Land use" can be defined as activities performed on the land surface to induce changes in land cover for different purposes, including social and economic purposes [37]. "Land cover" refers to the characteristics of the land surface, which can be defined by types, such as those of soil, terrain, water bodies, and buildings [38]. Land cover can be directly identified using remote sensing data, but additional information is required for land use classification [39,40]. Therefore, it is essential to primarily establish consistent criteria for delineating the entire national territory, which requires a multifaceted consideration of domestic and international factors.

Firstly, one can consider the availability of relevant national statistics and status of supporting spatial data. In South Korea, various spatial data have been constructed since 1995 by the National Spatial Data Infrastructure project [41]. Hence, it is crucial to ensure consistency between spatial data, national statistics, and international submissions. Specifically, it was necessary to verify the AFOLU sector's inventory against international



reports to ensure credibility [42]. In South Korea, the consistency of forest land and cropland categories reported to the FAO and UNEP must be maintained. Land use classification should correspond with international statistics, necessitating national-level, flexible, and clear criteria. If uniform standards are unattainable due to specific national contexts, similar to the USA and Japan, prioritizing land classification in data analysis is essential [18,43]. In South Korea, the definition of settlements' spatial extent varies by land use classification concept, necessitating government departments to develop a unified classification criterion through comprehensive discussions.

### 4.2. Construction of Activity Data

South Korea's Greenhouse Gas Inventory and Research Center (GIR) validates TACCC-related activity data across two key categories. Firstly, data from official national statistics must be compared with their original sources, whereas data from measurements or unofficial sources must include supporting evidence. Secondly, the suitability of the activity data for GHG inventory calculations in the sector, including its consideration of uncertainties, needs verification [42].

In this study, activity data construction followed the error range GL of GPG 2003-LULUCF and 2006 IPCC, employing a sampling method to calculate at the Tier 2a level as outlined in the 2006 IPCC GL. This approach is commonly applied in gathering activity data for the area estimation in Reducing Emission from Deforestation and Forest Degradation plus (REDD+) projects and the AFOLU sector [44]. Establishing activity data adhered to these criteria, achieving high precision, especially by employing high-resolution aerial orthoimages $25 \times 25$ cm$^2$ in size. Given that the biomass distribution within settlements is generally minimal relative to the low spatial resolution of remote sensing data, such data are unsuitable for generating national datasets [22,45,46]. Using high-resolution imagery for ongoing data collection offers insights into land use change dynamics and its associated human and environmental impacts [38,47–49]. The uncertainty in constructing activity data in this study can be assessed from two perspectives. Firstly, statistical calculations can be used to determine uncertainty based on the sample size and standard deviation of the survey data [50]. Secondly, uncertainty can be determined through the visual interpretation of remote sensing data to assess uncertainty in data attributes [51]. To address uncertainty in activity data, error matrices are constructed with additional field data. To precisely assess its data collection infrastructure's effectiveness, it is crucial to obtain comprehensive field survey data, which includes frequent updates for national application. Therefore, establishing baseline data to quantify information uncertainty is essential.

Demonstrating activity data accuracy is considered a mark of excellence [52]. The IPCC notes significant uncertainties in the AFOLU sector's GHG inventory, which impacts the TACCC principles [36]. The PA also requires parties to estimate and report their inventory uncertainties, methods, and assumptions [53]. This study was limited to data collection solely on the crown-cover area and did not cover all GHG inventory sources. Assumptions were made that withering organic material activity data is equal to crown-cover area, following the 2006 IPCC GL. It is important to differentiate between inorganic and organic soil activities, as such variations limit the applicability of remote sensing data. Therefore, to generate precise activity data, ongoing field surveys are indispensable.

To ensure the consistency of land use classification and avoid the duplication of activity data, the area was calculated by excluding activity data that overlap with DFTMs and SFMs. In 2019, there was a spatial overlap of 85,510.12 ha (42.6% of the total area) in settlements activity data, indicating significant variance based on land use classification criteria. Spatial overlaps varied according to the characteristics of land distribution in different regions. Significant duplication of activity data within the 19 categories in the CDMs of settlements, particularly in mixed-use areas, sports facilities, and site lands, was noted. Given these observations, departments involved in GHG inventory must achieve consensus on land use classifications.

This study identified regions with significant reductions in activity data in LS and increases in SS. This information can be used to manage carbon sinks in settlements from a policy perspective and to inform land use change management strategies. The activity data for 2000 and 2019 were constructed by land use change over 20 years, but the specific times of change were unclear. To conduct a detailed analysis of activity data and identify changes in location and time, shorter survey intervals are required [54]. Furthermore, it is crucial to understand changes in GHGs inventory, considering regional characteristics and the influence of policies [2,55,56].

*4.3. Evaluation of GHG Inventory Statistics*

Calculating the results revealed that settlements contributed 1.57% of the total $CO_2$ absorption in the AFOLU sector in 2019. After excluding for overlaps with DFTMs and SFMs, the figure was refined to 0.61%. These results stem from the decrease in $CO_2$ emissions due to the reduction of the spatial extent of settlements in LS and the diminished absorption in SS. Regionally, the inclusion or exclusion of overlapping areas significantly affected the $CO_2$ emission and absorption, attributed to variations in the proportion of LS reduction and the current crown-cover area percentage of SS, which differ by region. Understanding these regional differences is crucial for setting the spatial extent of GHGs inventory calculations within the settlement category. Moreover, the $CO_2$ absorption in settlement categories is relatively low compared to other land categories. Considering the continuous increase in settlement areas, calculations based on clear land use classifications are necessary.

To maintain the content of the current reporting of $CO_2$ absorption in the AFOLU sector and NDC achievement, delineating areas overlapping with settlements as forest or cropland for GHG inventory purposes may be beneficial. However, achieving TACCC in such land use classifications presents challenges. When prioritizing the estimation of forest land and cropland areas in overlapping regions, establishing a clear basis is essential. However, in South Korea, such groundwork for setting priorities is not currently being undertaken. Therefore, it is crucial to develop basis, which should then inform interdepartmental agreements and the reporting of GHGs inventories.

This study calculated carbon emission and absorption coefficients for evaluating GHG inventory statistics in LS using Tier 1 methodology, excluding FS (Tier 2), and assumed no $CO_2$ absorption settlement conversion. However, settlements initially result in vegetation loss but can evolve into urban areas offering carbon mitigation through afforestation [57–61]. Assessing $CO_2$ absorption after conversion demands detailed biomass, growth rates, and tailored emission and absorption coefficients for settlements. The gaps in data highlight the necessity for continuous data collection and the creation of specific coefficients suitable for Tier 2 calculations within settlement areas. Moreover, based on U.S. research [60], default absorption coefficients may not reflect actual absorption rates in different regions or settlement types, underscoring the need for country-specific coefficients reflective of local conditions [62,63]. City forests and roadside trees of South Korea show differences in carbon absorption per ha compared to the default coefficient in the 2006 IPCC GL Tier 2a [64–67]. However, this research, focusing solely on carbon absorption in small-scale areas at a single point in time, falls short of establishing national emission and absorption coefficients. Hence, there is a call for gathering detailed data on biomass, among other factors, within settlements to create country-specific coefficients aligned with the national context. Moreover, the study's GHG inventory for the AFOLU sector only accounted for biomass. Developing comprehensive activity data for these elements is crucial for more precise GHG inventory. The diversity of settlements and complexities in land ownership highlight the challenges of standardizing data collection, underlining the importance of adopting long-term strategies for gathering data on litter, dead trees, and soil carbon. This approach is vital for a comprehensive analysis of the GHG inventory dynamics of settlements.

## 5. Conclusions

The PA and compliance checks have made it important to calculate GHG inventory more scientifically and transparently based on the international TACCC principles. However, in the AFOLU sector of South Korea, five governing bodies are involved, and there is currently no agreed-upon standard for land use classification. Furthermore, the settlements category does not account for GHG inventory, resulting in limitations in securing TACCC in the AFOLU sector in South Korea. Therefore, this study aims to propose a methodology to construct activity data for estimating the GHG inventory of settlements and to calculate overlapping areas with other land categories. It aims to understand the changes in GHG inventory based on land use classification in the AFOLU sector and explore approaches to ensure TACCC based on this understanding.

For this purpose, this study first established settlement spatial extent using the CDMs and analyzed land use changes over a period of 20 years. The second step was constructing activity data within settlements using sampling methods based on changes in land use and analyzing the area of activity data with spatial overlap of lands according to the definition of other land categories. Third, the GHG inventory statistics for the settlements category were calculated using the activity data constructed and compared them with the statistics after excluding other land categories and overlapping areas.

According to the analysis, settlements accounted for approximately 11% of the total land area of South Korea. The area of LS was found to be 32%, and the area of SS was found to be 68% due to land use change in 20 years; the historical area of each activity data was 0.23 million ha for LS and 0.03 million ha for SS. The current area was 0.07 million for LS and 0.13 million for SS. The area overlapping between CDMs and DFTMs was 10% of the settlement area of CDMs, whereas SFMs showed an overlap of approximately 6% with the settlement area. Additionally, in 2019, the settlement area consisted of 18.5% activity data based on CDMs and 12.66% activity data after excluding overlapping areas of DFTMs and SFMs. Based on these results, the GHG inventory statistics for the category settlement showed an absorption of 1.57% of $CO_2$ emissions and absorptions reported in South Korea's AFOLU sector as of 2022. Excluding overlapping areas, the absorption was 0.61% of the total $CO_2$ emissions and absorptions. Therefore, discussions are needed regarding the spatial extent of settlement in South Korea, including that for other land use categories. Assessing the expected GHG inventory statistics based on land use classification will be necessary for setting the direction and conducting progress checks on achieving national NDCs in the future.

**Author Contributions:** Conceptualization, S.-E.C. and W.-K.L.; methodology, S.-E.C., M.K., Y.S., K.-H.L. and S.-W.J.; software, S.-E.C.; validation, M.K., Y.S., K.-H.L., S.-W.J. and W.-K.L.; formal analysis, S.-E.C.; investigation, S.-E.C. and W.K.; resources, S.-E.C. and W.K.; data curation, S.-E.C.; writing—original draft preparation, S.-E.C.; writing—review and editing, S.-E.C., W.K. and S.-J.L.; supervision, W.-K.L.; project administration, W.-K.L.; funding acquisition, W.-K.L. All authors have read and agreed to the published version of the manuscript.

**Funding:** This research was funded by National Research Foundation of Korea (NRF-2021R1A6A1A10045235), Ministry of Land, Infrastructure and Transport/Korea Agency for Infrastructure Technology Advancement (23UMRG-B158194-04), National Institute of Forest Science (FM0200-2023-01-2023).

**Data Availability Statement:** The data presented in this study are available in article.

**Conflicts of Interest:** The authors declare no conflicts of interest.

**Abbreviations**

| | |
|---|---|
| AFOLU | Agriculture, Forestry and Other Land Use |
| BTR | Biennial transparency reports |
| BUR | Biennial update reports |
| CDMs | Cadastral maps |
| DFTMs | Digital forest type maps |
| ETF | Enhanced Transparency Framework |
| GHG | Greenhouse gas |
| GIR | Greenhouse Gas Inventory and Research Center |
| GL | Guidelines |
| IPCC | Intergovernmental Panel on Climate Change |
| IPPU | Industrial Processes and Product Use |
| LS | Land converted to settlements |
| LULUCF | Land use, land use change, and forestry |
| MRV | Monitoring–reporting–verification |
| NDCs | Nationally determined contributions |
| NIR | National inventory reports |
| PA | Paris Agreement |
| REDD+ | Reducing Emission from Deforestation and Forest Degradation plus |
| SFMs | Smart farm maps |
| SS | Settlements remaining settlements |
| TACCC | Transparency, accuracy, completeness, comparability, consistency |
| UNFCCC | United Nations Framework Convention on Climate Change |

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
