# Peer review of "Development of Activity Data for Greenhouse Gas Inventory in Settlements in South Korea"

_land, doi:10.3390/land13040497_

Round 1

Reviewer 1 Report

Comments and Suggestions for Authors

Dear authors

The article summarizes a study on land use and CO2 adsorption. The given information, however, does not bring anything research that needs to be published in research journals. I'm sorry, but the given data is useful for the given country, suitable for presentation, but not suitable for publication in a journal whose content is research and development. I don't see anything "new or novelty" anywhere, which is the important for research articles.

I am very sorry, but this is not a research paper in the true sense of the word. It is therefore not possible to comment on the content and results from the research point of view.

Best regards

Author Response

We are very appreciate for dedicating your valuable time to reviewing the manuscript.

We apologize for the understanding the manuscript because our explanation is insufficient. The purpose of this study is to propose methodologies for constructing activity data for the settlements category, addressing the challenges encountered by existing non-Annex I countries, required to submit Biennial Transparency Reports (BTR) under the Paris Agreement, when estimating greenhouse gas inventories in the AFOLU sector. Additionally, the study aims to suggest solutions for issues related to overlapping activity data due to land-use classification settings.  

Therefore, the paper has been sufficiently revised to emphasize these points. Based on the research findings, setting directions for land-use classification for greenhouse gas inventory reporting in various countries and identifying and addressing potential issues arising from national status could be facilitated.

Reviewer 2 Report

Comments and Suggestions for Authors

General comment: The authors investigate the quality of data bases for a GHG inventory in South Korea. This topic is interesting and deserves publication in Land. However, after reading the manuscript it is unclear if the new methods applied to calculate the GHG inventory are beneficial or if the different methods just provide different results, showing the uncertainty in the calculation of the GHG inventory. This aspect should be added in the Discussion and Conclusions Sections. Most figures need improvement (see below).

Provide a list of abbreviations before the introduction.

Detailed comments:

Line 35: add a definition of the Paris agreement (main items)

Line 114: for consistent criteria

Line 128: settlements

Line 142: centered at 37°N and 127°E ?

Line 149: leading to

Line 151: which houses 91.9% of South Korea's total population: Does this mean that 91,9% of the population lives in urban areas or in Seoul alone (then "which houses" would be correct; otherwise "which house" is correct)

Line 169: cadatral maps

Figure 1: To which of the sub-figures does the legend apply? Figure is not clear at all. More explanation is needed.

Line 180: delete "another"

Line 260: either "guideline categorizes" or "guidelines categorize"

Line 285: Eq. 2

Line 286: second Ai,j is probably Vi,j

Lines 307-309: Explain the difference between the total settlement area, LS and SS (the latter two comprise only about 10 % of the total settlement area)

Figure 3: explain legend (FS, CS and so on)

Line 330: 6.7 ha? (not 6.7 million ha?)

Lines 381-382: Reference to Fig. 5 is missing; more interpretation of Fig. 5 is needed.

Figs. 4-6: legend is too small to be readable; improve figures

Line 477: replace "lands category" by "land categories"

Line 498: delete "a" before consistent criteria

Line 518: establish criteria (the singular is "criterion")

Line 574: better: changes in location and time

Line 625: of a criterion or of criteria

Line 651: to be deleted

Comments on the Quality of English Language

English is fine, only minor editing is required.

Author Response

We are very appreciate for dedicating your valuable time to reviewing the manuscript.

We have made every effort to respond to your comments.

The revised content has been documented in a Word file and uploaded. The modified sections are highlighted in blue in the resubmitted manuscript.

Best regards,

Reviewer 3 Report

Comments and Suggestions for Authors

Dear Authors,

The research in the article “Development of Activity Data for Greenhouse Gas Inventory in Settlements in South Korea” aimed to propose a method for examining the settlement spatial extent and constructing activity data to estimate GHG emissions and absorption as a pilot calculation, and to provide data for land-use classification.

The introduction provides sufficient background with relevant references. The description of the methodology is detailed. The results are presented clearly and legibly. The discussion is appropriate.

The article is overall well-written, and easy to read. I do not have major criticisms of this article. 

Author Response

We are very appreciate for dedicating your valuable time to reviewing the manuscript.

The manuscript has been revised comprehensively once again to emphasize the purpose and applicability of the research. The revised sections can be reviewed in the resubmitted manuscript.

We would like to express our gratitude for your participation in the review of this paper.

Best regards

Round 2

Reviewer 1 Report

Comments and Suggestions for Authors

Dear authors

Thank you very much for adding to the article and I fully agree with the statement. I clearly see the novelty and agree to the publication of the given article.

Best regards